# Preparation of Colored Microcapsule Phase Change Materials with Colored SiO_2_ Shell for Thermal Energy Storage and Their Application in Latex Paint Coating

**DOI:** 10.3390/ma14144012

**Published:** 2021-07-18

**Authors:** Enpei Ma, Zhenghuang Wei, Cheng Lian, Yinping Zhou, Shichang Gan, Bin Xu

**Affiliations:** 1College of Materials Science and Engineering, Zhejiang University of Technology, Hangzhou 310000, China; m13984303134_1@163.com (E.M.); wzh01020@163.com (Z.W.); zhouyinp326@163.com (Y.Z.); gsc1509035997@163.com (S.G.); 2Siao Holdings Co., Ltd., Linan 311300, China; whlc168@163.com

**Keywords:** phase change materials, microcapsule, colored SiO_2_ shell, solar thermal energy storage, latex paint coating

## Abstract

This article reports the design and manufacture of colored microcapsules with specific functions and their application in architectural interior wall coating. Utilizing reactive dyes grafted SiO_2_ shell to encapsulate paraffin through interfacial polymerization and chemical grafting methods, this experiment successfully synthesized paraffin@SiO_2_ colored microcapsules. The observations of surface morphology demonstrated that the colored microcapsules had a regular spherical morphology and a well-defined core-shell structure. The analysis of XRD and FT-IR confirmed the presence of amorphous SiO_2_ shell and the grafting reactive dyes, and the paraffin possessed high crystallinity. Compared with pristine paraffin, the thermal conductivity of paraffin@SiO_2_ colored microcapsules was significantly enhanced. The results of DSC revealed that the paraffin@SiO_2_ colored microcapsules performed high encapsulation efficiency and desirable latent heat storage capability. Besides, the examinations of UV-vis and TGA showed that the paraffin@SiO_2_ colored microcapsules exhibited good thermal reliability, thermal stability, and UV protection property. The analysis of infrared imaging indicated that the prepared latex paint exhibited remarkable temperature-regulated property. Compared with normal interior wall coatings, the temperature was reduced by about 2.5 °C. With such incomparable features, the paraffin@SiO_2_ colored microcapsules not only appeared well in their solar thermal energy storage and temperature-regulated property, but also make the colored latex paint coating have superb colored fixing capabilities.

## 1. Introduction

In recent years, numerous researchers have developed many building energy-saving technologies, in which the application of thermal energy storage technology in building materials has become the hotspot of research and attention [1,2,3]. Different from previous thermal energy storage materials, phase change materials (PCMs), as latent heat storage materials, can maintain their temperature within a certain range by absorbing or releasing latent heat. Therefore, PCMs are considered to be recyclable clean energy materials with high energy conversion efficiency. In addition, it can be acknowledged from public research that there are many organic or inorganic materials that can be used as solid–liquid PCMs, including organic fatty acids and paraffins, inorganic salt hydrates and esters [4,5,6]. Among the various PCMs, paraffin has attracted more and more attention due to its ideal properties (such as stable chemical properties, wide melting point range, no segregation tendency, high latent heat, and reasonable price) [7,8,9]. Moreover, because 28 °C is suitable for human comfort, industrial-grade paraffin is widely favored in residential construction applications [10]. Nevertheless, thermal conductivity and leakage issues limit the compatibility of paraffin in building materials.

To overcome the leakage issue and increase the heat transfer area of the solid–liquid PCMs in the liquid state, microencapsulation technology can be used to support the PCMs in the packaging container, so that the PCMs maintain a certain shape in both the liquid state and the solid state. Various methods have been employed for the encapsulation of PCMs through complex coacervation [11], interfacial polymerization [12], suspension polymerization [13], and in-situ polymerization [14]. Most of the reported researches have used polymers as packaging materials, such as urea-formaldehyde resin [15], poly(methyl methacrylate) [16], melamine–formaldehyde resin [17], and even bio-based materials such as silk fibroin and gelatin/gum Arabic [18]. The micro-encapsulation of PCMs droplets by these polymer shells can prevent the PCMs from leaking, and such a shell layer also has the required mechanical strength to hold the structural stability of microcapsules. However, compared with metals and traditional inorganic materials, the inherent low thermal conductivity and flammability of polymer materials will significantly affect their thermal response and thermal conversion performance, resulting in hysteresis in the thermal adjustment process. Apart from being superior to polymeric materials in terms of non-combustibility and thermal conductivity, inorganic materials also have mechanical strength and thermal and chemical stability [19]. The combination of organic PCMs and inorganic shell materials can improve the energy storage capacity and thermal performance. As a result, there has been growing interest in synthetic technologies for the microencapsulation of inorganic substances in recent years. For example, various inorganic shells, such as CaCO_3_ [20], Al(OH)_3_ [21], ZnO [22], TiO_2_ [23], and Cu_2_O [24], have been successfully used as packaging materials to design a series of microencapsulated PCMs. The resulting microcapsules exhibit good phase change thermal performance due to the high thermal conductivity of the inorganic shell, and the rigid and compact inorganic shell also provides better thermal stability, cycle durability, and better package and permeation resistance, acting more effectively than polymers as a reliable barrier to protect the PCMs from negative interactions with the surroundings.

Interestingly, by choosing a specific inorganic shell to cover the organic PCM, it can be found that certain chemical or physical functions of the specific inorganic shell can be transferred to the final microcapsules. It is undoubted that SiO_2_ has the advantages of non-toxicity, stable structure, clear surface characteristics, and simple preparation [10,25,26,27]. In addition, it is commonly known that the synthetic SiO_2_ shell has poor compatibility with various hydrophobic polymer materials due to its high specific surface area and hydrophilicity. In recent years, there have been lots of researchers focusing on microencapsulated PCMs of polymer-silica hybrid shells. Chang et al. [28] synthesized microencapsulated PCMs with a PMMA–silica hybrid shell via a sol–gel process. Li et al. [29] prepared microencapsulated PCMs with hybrid shells through the polymerization of two alkoxysilanes. It is worth mentioning that much literature reports research of microcapsule PCMs in the field of building wall materials. For example, Karkri et al. [30] successfully manufactured microencapsulated paraffin/stucco building materials to store thermal energy through compression molding. It can be concluded that the prepared stucco/microencapsulated paraffin composite materials, as a thermal energy storage material, are expected to reduce energy consumption in construction applications and have broad prospects. Although these phase change microcapsules endow the wall coating with heat storage properties, most of them have a single color, which is difficult to meet the aesthetic needs of modern people. Organic reactive dyes are the main colorants used in the dyeing of cotton fabrics, which have excellent sunlight and weather resistance and can be used in the field of building walls [31,32,33]. For example, Lei et al. [34] studied the effect of utilizing colored cooling paint and PCMs as an auxiliary cooling strategy in reducing the cooling load of tropical buildings through experiments and numerical researches. The results demonstrated that the color coating and PCMs are two complementary passive cooling strategies to absorb the conductive heat that cannot be handled by the color cold coating. Soudian et al. [35] researched and explored the potential of a new type of cementitious material combined with PCMs and TC colored coatings as facade facing materials to control solar energy and heat load under different seasonal conditions. The results indicated that the sample has the potential to reduce high-temperature solar reflectance and reduce temperature fluctuations. However, The PCMs and colored coatings mentioned above are prepared by physical adsorption or Coulomb force methods to prepare colored thermal insulation coatings, its solvent resistance, color characteristics and washing fastness are all unsatisfactory. After long-term corrosion in the environment, it is easy to fall off and fade, and it is difficult to achieve the purpose of color fixation. As a consequence, fixing the colorant permanently on the SiO_2_ matrix is a way to overcome these shortcomings. However, it is indicated in the published literature that SiO_2_ cannot directly target reactive dyes [36,37,38,39,40]. Accordingly, an attempt could be inspired to introduce some groups on the shell material of the microcapsules, and SiO_2_ shell could be chemically grafted and immobilized with reactive dyes to prepare phase change color microcapsules with super firm color fixing ability to increase their color diversity. Therefore, the organic modification of silica shell materials is beneficial to improve the compatibility of microcapsules with a variety of hydrophobic polymer substrates, as well as to more firmly anchor reactive dyes.

Herein, we develop a new strategy to fabricate a new type of colored microcapsule PCMs with thermal energy storage and ultraviolet (UV) protection properties. On the basis of the best morphology silica inorganic wall material microcapsules, the active hydroxyl groups on the surface of the paraffin@SiO_2_ phase change microcapsules are modified with a silane coupling agent through triaminopropyltriethylsilane (KH550). Thus, synthesize organic modified paraffin @SiO_2_ phase change microcapsules containing amino groups on the surface. Then, carry out a nucleophilic substitution reaction between the introduced amino group and the active Cl atom of the organic reactive dye. After testing and analysis, it was found that the colored microcapsules not only showed good thermal physical properties, thermal conductivity, and thermal stability, but also showed bright colors. Therefore, the application of paraffin@SiO_2_ colored microcapsules to building materials can not only alleviate energy problems and improve thermal comfort, but also greatly increase the color diversity of interior wall coatings.

## 2. Material and Reagents

The paraffin used as a latent heat storage material with a melting point about 28 °C was supplied by Rubitherm (Shijiazhuang, China) PCMS Co., Ltd.; the properties of the paraffin are shown in Table 1. Tetraethyl orthosilicate (TEOS) was supplied by Shanghai Lingfeng Chemical Reagent Co., Ltd. (Shanghai, China). Cetyl Trimethyl Ammonium Bromide (CTAB) was provided by Shanghai Bio Biotech Co., Ltd. (Shanghai, China). 3-Aminopropyltriethoxysilane (KH550) was purchased from Shanghai Aladdin Biochemical Technology Co., Ltd. (Shanghai, China). Hydrochloric acid (HCl, 37.5 wt.%) was supplied by Xiqiao Science and Technology Co. (Shanghai, China). Ammonia water was supplied by Hangzhou Minshan Fine Chemical Co., Ltd. (Hangzhou, China). Ltd. Formamide was provided by Shanghai Aladdin Biochemical Technology Co., Ltd. (Shanghai, China), and anhydrous ethanol was provided by Anhui Ante Food Co., Ltd. (Suzhou, China). The reactive dyes were purchased from Jinan Haoxing Chemical Co., Ltd. (Jinan, China), and the structure formula is shown in Figure 1. All reagents are analytical grade and are used as received without further purification. The parameters of the experiment and equipment are shown in Table 2 and Table 3.

### 2.1. Methods

#### 2.1.1. Synthesis of the Paraffin@SiO_2_ Microcapsules

The paraffin@SiO_2_ microcapsules were prepared by referring to the previous literature [41]. The microencapsulated paraffin with SiO_2_ shell was synthesized through interfacial polymerization in a non-aqueous emulsion template system. TEOS was used as a silicon source. CTAB is used as a cationic surfactant and hydrochloric acid as a catalyst. A typical synthesis process is as follows. In a beaker, 10 g of paraffin and 10 g of TEOS were mixed and magnetically stirred at 45 °C for 30 min. 1.3 g CTAB was dissolved in 100 mL formamide and mechanically stirred in a three-neck round bottom flask for 30 min to obtain a homogeneous emulsion. Subsequently, the paraffin/TEOS composite oil phase was transferred to the three-neck round flask and agitated with 5 h to acquire a mixed non-aqueous oil-in-water (O/W) emulsion. After that, the aqueous solution of hydrochloric acid (100 mL, 1.8 mol/L) was added slowly to the above-prepared emulsion and slowly stirred for 2 h. Then, the mixture was continuously stirred for 4 h. After ageing at 50 °C for 18 h, the resulting solid–liquid mixture was centrifuged, washed with n-hexane and warm water 3 to 5 times, and dried in a 50 °C drying oven for 12 h to obtain paraffin@SiO_2_ microcapsules, waiting for further preparation.

#### 2.1.2. Synthesis of Colored Microcapsules

Reactive dyes were grafted onto the surface of paraffin@SiO_2_ microcapsules by chemical method; the process is as follows: 10 g paraffin@SiO_2_ microcapsules, 100 mL absolute ethanol and 10 mL distilled water were mixed in a beaker and sonicated at 40 °C for 30 min. Subsequently, the paraffin@SiO_2_ microcapsules dispersed phase was transferred to a three-necked flask, and 5 g KH550 added to 40 mL absolute ethanol dropwise until the dispersion was uniform; after that, the resulting solution was dropped dropwise into the dispersed phase of paraffin@SiO_2_ microcapsules and reacted for 2 h. In succession, 15 mL ammonia and 1 g of X-Br were added to the above dispersed phase and slowly stirred for 2 h. Finally, the resulting microcapsule samples were centrifuged, washed in absolute ethanol and distilled water, and dried in a 50 °C drying oven for further measurement.

#### 2.1.3. Preparation of Temperature-Regulating Coating

As a comparison, 15% colored SiO_2_ mixed with reactive dyes and 15% paraffin@SiO_2_ colored microcapsules was added to a latex paint coating formula (65% silicone acrylic emulsion, 1.5% ethylene glycol, alcohol ester 2.0%, bactericide 0.2%, thickener 0.6%, water) and mixed evenly by magnetic stirring at 1000 rpm for 30 min to obtain a normal colored coating: respectively, A and colored coating B containing particular functional microcapsules. Subsequently, coating A and coating B were respectively coated on a highly transparent silica glass substrate with a size of 5 × 5 mm, with a coating thickness of 0.5 mm. Then, this was cured at room temperature for 5 days, and the heat preservation performance of the coatings was tested after it dried.

### 2.2. Characterizations

The morphological characteristics of the microcapsules were observed by a field emission scanning electron microscope (FE-SEM, ΣIGMA300) manufactured by Zeiss, Germany. Then, Cu-Kα radiation (λ = 0.15406 nm) was used to obtain the crystal characteristics of the microcapsule samples by X-ray diffraction (XRD, X’PERT PRO, Almelo, Holland). The Fourier-transform infrared (FT-IR) spectra of the microcapsule samples were studied through a Fourier-transform infrared spectrometer (Thermo Nicolet 6700, Waltham, MA, USA) applying the KBr pellet method.

The thermal performance of the paraffin and the microcapsule samples was determined by a differential scanning calorimeter (DSC, STARE system, Mettler Toledo, Zurich, Switzerland). The detailed DSC test conditions are that 7 ± 1 mg of each sample is sealed in aluminum at a heating rate of 10 °C/min and a flow rate of 50 mL/min under a constant nitrogen flow. At a heating rate of 10 °C/min from room temperature to 700 °C, and a flow rate of 50 mL/min, the thermal stability of the colored microcapsules and paraffin samples was studied under nitro conditions applying a thermogravimetric analyzer (TGA, Q5000, TA Instruments-Waters LLC, New Castle, UK). The thermal conductivity of the microcapsule samples and paraffin was obtained by the transient hot wire method applying the Xia tech hot plate thermal conductivity meter (TC-3000). In order to measure the thermal conductivity, the microcapsule sample was pressed into two rectangular parallelepipeds with a length of 3 cm, a width of 2 cm, and a height of 1 cm, then the sheet-shaped thermal conductivity sensor flattened with a mold. Between the two sample cuboids, the weight of the top of the cuboid is 1 kg, and the inspection result is obtained by the average of five measurements.

Additionally, the paraffin and microcapsule samples were analyzed by ultraviolet-visible spectroscopy (UV-visible spectrophotometer (Lamda 750 s, Perkinelmer, Waltham, MA, USA). In the light absorption performance test in this article, the same mass of paraffin and paraffin@SiO_2_ colored microcapsules uniformly dispersed in deionized water were weighed for testing. The measured spectrum ranges from 250 nm to 400 nm. As shown in Figure 2, the solar energy thermal storage capacity of the pristine paraffin and microcapsule samples is tested through a self-assembled test device, which includes a transparent container, a reflective heat insulation system, a simulated light source, and a data acquisition system. The data acquisition system is composed of computer, data collector, and thermocouple. In the photo-thermal temperature test system, two 250 W solar simulation light sources are utilized to simulate solar radiation, and the total irradiance is 500 W·m^−^^2^. When the solar simulation light source is turned on, the heat storage performance of the sample is measured. When the temperature of the sample reaches the phase transition temperature, the solar simulation light source is turned off. Then, the sample is rapidly cooled to room temperature, during which the sample has an exothermic process. The temperature change of the samples during these periods is automatically recorded by the data collector. When testing, the time interval for data collection is 1 s, and the accuracy of the thermocouple is ±0.1 °C. The microcapsule sample and X-Br are tested on the KONICA MINOLTA CM-3500 d spectrophotometer (Keshengxing Instrument Co., Ltd., HangZhou, China) to test the colored parameters to obtain the L*, a*, and b* values. The test refers to the CIELAB uniform L* a* b* color space recommended by the International Commission on Illumination (CIE). The color space is composed of a lightness factor L* and two chromaticity factors a* and b*.

## 3. Results and Discussion

### 3.1. Synthetic Strategy and Formation Mechanisms

The paraffin@SiO_2_ colored microcapsules are synthesized by a two-step method. Figure 3 is a synthetic mechanism diagram of encapsulated paraffin with colored SiO_2_ shell. First, as shown in the Figure 3 in a non-aqueous emulsion system with CTAB as emulsifier, the paraffin@SiO_2_ microcapsules were prepared by the interfacial polycondensation method of the silica precursor on the surface of the emulsified droplets. In this paper, formamide solution is utilized as the template medium, which can alleviate the rapid hydrolysis rate of TEOS in water as a non-aqueous system. Among them, due to the emulsification of emulsifier CTAB, a surfactant film is formed on the surface of the paraffin/TEOS oil phase, which resulted in a stable oil-in-water emulsion. Next, these silica precursors are attracted to the micelle surface through the hydrogen bonding mutual attraction between the silica precursor and the hydrophilic segment of the surfactant. Due to the hydrolysis and polycondensation rate of silica precursor in water being extremely fast, it is too late to assemble on the surface of the emulsion micelle droplets, but a large number of silicon self-polymer particles are formed, which makes it difficult for the core material to be effectively coated forming microencapsulated PCMs; so this article chose to use formamide solution as the template medium, which alleviates the rapid hydrolysis rate of TEOS in water as a non-aqueous medium. Subsequently, a certain amount of HCl aqueous solution is dropped dropwise to promote the interfacial polycondensation of silica precursor, then it is bonded to the surface of the paraffin droplet through hydrogen bonding, and the paraffin core is wrapped in the silica shell. In addition, in the process of synthesizing microcapsules, it is verified that when the mass ratio of paraffin/TEOS is designed as 50/50, the obtained microcapsules could have a suitable thickness and better encapsulation efficiency. Subsequently, further chemical grafting is carried out, and when an appropriate amount of modifier (KH550) is added to the dispersed phase of microcapsules, a numerous amount of hydroxyl groups will be generated through the hydrolysis reaction, which continue to conduct polycondensation reaction with the hydroxyl groups of inorganic SiO_2_ shell, making KH550 anchor on the its surface and successfully introducing -NH_2_ to obtain modified SiO_2_ microcapsules, and further, adding organic reactive dyes under alkaline atmosphere. Through regulating the mass ratio of KH550 and the microcapsules, the reactive Cl atoms of amino groups and organic reactive dyes are paraffin@SiO_2_ microcapsules modified by KH550 to undergo nucleophilic adsorption by van, and reactive dye molecules are attracted and fixed to the SiO_2_ shell of microcapsules. In our current work, the paraffin@SiO_2_ colored microcapsules were prepared from a dichloro-s-triazine-type reactive dye (C.I. Reactive Blue 4), and the apparent appearance of the obtained sample is shown in the Figure 3. Specifically, by virtue of the nucleophilic substitution reaction occurring between the active Cl atom of reactive dye molecules and the amino groups on the modified SiO_2_ shell, the dye molecules are tightly anchored on the SiO_2_ shell, resulting in the dye molecules being tightly fixed on the SiO_2_ shell, thus showing bright color.

### 3.2. Chemistry and Crystal Structure of Microcapsules

Figure 4 is the SEM image of paraffin@SiO_2_ microcapsules and paraffin@SiO_2_ colored microcapsules, in which it can be exhibited that the microcapsule samples perform a regular spherical shape with the diameter sizes of 1–5 µm. From the magnified images c and d of the microcapsules, it can be seen that the surface of c is smoother than that of d, and there are some protruding and transparent attachments in d, which may be caused by the grafting of reactive dyes onto the SiO_2_ shell. Figure 5 shows SEM pictures of two microcapsule samples deliberately ground and damaged, indicating the microstructure of these damaged microcapsules in detail, in which it can be clearly found that the microcapsules and the obtained colored microcapsules have a typical core-shell structure.

### 3.3. Chemical Composition and Crystal Structure

The crystallinity of paraffin@SiO_2_ microcapsules and paraffin@SiO_2_ colored microcapsules was examined by XRD, which are exhibited in Figure 6. As can be revealed in the Figure 6, five diffraction peaks of both paraffin@SiO_2_ microcapsules and paraffin@SiO_2_ colored microcapsules appear at 2θ = 8.460°, 12.371°, 20.560°, 22.005°, and 24.366° corresponding to (001), (110), (111), (230), and (401) crystal faces of paraffin, while the broad peak of 2θ at 23° indicates the characteristic peak of amorphous SiO_2_, and no other new diffraction peaks appear, which means that no chemical interaction occurs between pristine paraffin and SiO_2_, which maintains the thermal stable properties of pristine paraffin.

The FT-IR spectrogram of pristine paraffin, paraffin@SiO_2_ microcapsules, paraffin@SiO_2_ modified microcapsules, and paraffin@SiO_2_ colored microcapsules is presented in Figure 7. As can be seen from the spectrogram of pristine paraffin, the absorption peaks at 2919 cm^−1^ and 2848 cm^−1^ are caused by the stretching vibration of the methyl and methylene of paraffin, respectively. Furthermore, the peak at 677 cm^−1^ represents the in-plane rocking vibration of the -CH_2_ group. The peaks at 1736 cm^−1^ and 1464 cm^−1^ are assigned to C = O stretching vibration and C-H bending vibration, which are typical paraffin absorption peaks, and the corresponding peaks are found in the microcapsules PCMs b, c, and d, indicating the presence of paraffin in the microcapsule samples. Moreover, it should be noticed that three microcapsules PCMs all show prominent absorption peaks at 1050 cm^−1^, 1020 cm^−1^, and 1030 cm^−1^, which are broad and strong, and they are Si-O-Si antisymmetric stretching vibration absorption peaks. The absorption peaks at 773 cm^−1^, 775 cm^−1^, and 783 cm^−1^ in the three curves of b, c, and d are ascribed to the bending vibration of Si-O-Si, revealing that the SiO_2_ shell exists in the microcapsules. The corresponding absorption peaks of curve b at 966 cm^−1^ and 3440 cm^−1^ are the symmetric and antisymmetric absorption peaks of Si-OH, and do not appear in c and d. This is because there are a large number of hydroxyl groups on the SiO_2_ shell of the paraffin@SiO_2_ microcapsules. After adding KH550, the hydroxyl groups on the SiO_2_ shell and the O-H produced by the hydrolysis of KH550 condense and form more Si-O-Si bonds, and the hydroxyl groups disappear, indicating that KH550 has been successfully connected to the SiO_2_ shell of the paraffin @SiO_2_ microcapsules. Besides, the peak of curve c at 1651.89 cm^−1^ is the characteristic bending peak of N-H. It is interesting to find that there is a broad peak at 3400 cm^−1^ in d again. This can be explained by the following. On the one hand, it can be the characteristic absorption peak of N-H in reactive dye. On the other hand, it can also be the peak formed by the nucleophilic substitution between the amino group in KH550 and the Cl atom of the reactive dye. In addition, the absorption peak at 2358 cm^−1^ is the characteristic absorption peak of the cyano group (-CN) of the reactive dye, which further demonstrates that the reactive dye has been successfully grafted to the surface of the paraffin@SiO_2_ modified microcapsules.

### 3.4. Thermal Conductivity

Thermal conductivity is considered to be a crucial parameter for thermal energy storage and thermal regulation, because it affects the thermal response speed of latent heat storage and release. In addition, the thermal conductivity of conventional polymer materials is in the range of 0.08–0.25 W·m^−1^·K^−1^. The results obtained at room temperature are presented in Figure 8. The thermal conductivity of paraffin@SiO_2_ microcapsules and paraffin@SiO_2_ colored microcapsules are 0.6215 W^−1^·K^−1^ and 0.7011 W^−1^·K^−1^, respectively. Compared with pristine paraffin (0.2619 W^−1^·K^−1^), it enhanced to 57% and 62%, which is significantly higher than that of polymer shell and paraffin core. There is no doubt that the encapsulation of PCMs by inorganic materials makes the microcapsule samples more sensitive to environmental temperature. In addition, after silane hydrolysis and polycondensation, the shell thickness of inorganic SiO_2_ is further increased, so that the colored microcapsule PCMs reaches a faster thermal response.

### 3.5. Phase Transition Characteristics and Thermal Performance Analysis

Generally, the phase transition temperature range of PCM is within the human body comfortable temperature range of 19–26 °C in the field of building energy saving. In addition, it is generally accepted that the latent heat of PCMs is one of the most significant factors affecting working performance evaluating the phase change behavior of the prepared microcapsule samples. The phase transition behavior of pristine paraffin and microcapsule samples can be investigated by DSC, and their DSC curve and related phase transition parameters are showed as Figure 9 and Table 4. The curve in the lower half of the graph is the temperature-rise period; the microcapsule PCMs absorb heat to generate the enthalpy of fusion; the upper half of the curve is the temperature-fall period; the microcapsules release latent heat to generate the crystallization enthalpy. Although all the microcapsule samples exhibited DSC spectra similar to pristine paraffin, the amplitude of the crystallization and melting peaks were significantly reduced. That is because in the temperature range measured by DSC, the inert materials SiO_2_ and dye have never undergone phase change. Therefore, the loading of paraffin in the microcapsules determines the latent heat of these microcapsule samples. Compared with paraffin, the crystallization temperature (T_c_) and melting temperature (T_m_) of the microcapsule samples decreased slightly. The result may be attributed to the crystal limitation of the narrow space in the microcapsule, which restricts the movement of paraffin molecules. On the other hand, the inner of the SiO_2_ shell provides heterogeneous nucleation points, which causes the phase change of the microcapsule PCMs at a lower temperature. What is more, the encapsulation efficiency (*E_en_*) and the encapsulation efficiency (*E_es_*) of the microcapsule samples can be calculated by the following equation:(1)$$E_{en}=\frac{\Delta H_{m,micro}}{\Delta H_{m,PCM}}\times 100\%$$
(2)$$E_{es}=\frac{\Delta H_{m,micro}+\Delta H_{f,micro}}{\Delta H_{m,PCM}+\Delta H_{f,PCM}}\times 100\%$$
where Δ*H_m,micro_* and Δ*H_f,micro_*, respectively, represent the latent heat enthalpy of melting and crystallization of the microencapsulated composite material. Generally, encapsulation efficiency and energy storage efficiency are deemed to be two major characteristic parameters describing the phase change performance of microencapsulated PCMs. Finally, calculated by formula, the encapsulation efficiency of paraffin@SiO_2_ microcapsule is 38%, and of paraffin@SiO_2_ colored microcapsules is 30%.

### 3.6. Temperature-Regulating Performance and Solar Energy Thermal Storage Capacity

As depicted in Figure 1, the homemade photo-thermal apparatus is applied to evaluate the solar energy storage efficiency and thermal regulation capability of the microcapsule samples through a constant ambient temperature. The apparatus can utilize artificial solar light sources to simulate the solar energy storage and release process of PCMs. It can be found out from the Figure 10a that under the irradiation of the simulated light source, the temperature of the paraffin and the two microcapsules gradually rises along with the increase of the lighting time; then, a distinct temperature plateau region appears within a temperature span of 25–31 °C, which is attributed to the hysteresis of the temperature rise led by the latent heat storage of the core paraffin in the microcapsule samples during the melting process. Complementarily, the phase change end temperatures of paraffin, paraffin@SiO_2_ microcapsules, and paraffin@SiO_2_ colored microcapsules are 30 °C, 29.1 °C, and 27.9 °C, respectively, which are basically unanimous to the results of DSC. Furthermore, it is worth noting that compared with pristine paraffin, the microcapsule samples have a much higher thermal conductivity, so it has a faster temperature rise under the same irradiation time, and it achieves a higher temperature after the illumination finishing. Similarly, the paraffin and microcapsule samples are naturally cooled at 20 °C ambient temperature after heating to 50 °C, which is similar to the photothermal storage process. It is also interestingly observed from the Figure 10b that the temperature hysteresis leads to the obvious plateau area in the temperature range of 28–22 °C because of the release of latent heat during the crystallization process of paraffin. On the contrary, the SiO_2_/dye composite does not appear to be such a constant temperature platform in the time-temperature curve, because it cannot produce a phase change to store solar energy.

### 3.7. UV Protection Property and Thermal Stability

Disperse 0.5 g microcapsule samples and reactive dye X-Br in 100 mL ethanol solution, respectively. A series of color parameters of the resulting samples is listed in Table 5. As shown in Table 5, the b value in the color coordinates L*, a*, b* is negative, indicating that the colored microcapsule PCMs and X-BR are blue, while the three color coordinates of the paraffin@SiO_2_ microcapsule samples are all near 0, and its L* value is almost close to 100, indicating that it is white.

The ultraviolet-visible spectrophotometric test was carried out to examine the UV protection performance of the microcapsules. The UV-vis absorption spectra of pristine paraffin, paraffin@SiO_2_ microcapsules, paraffin@SiO_2_ colored microcapsules, and SiO_2_ are displayed in Figure 11a. The increase in absorbance in UV region (both UVB (290–320 nm) and UVA regions (320–400 nm) indicates the increase of UV protection quality. It is interesting to indicate that the spectrum of pristine paraffin has no absorption in the ultraviolet region, while the ultraviolet absorption of the microcapsule samples increased significantly because of the increase in the thickness of SiO_2_ shell in the microcapsule samples and the numerous ultraviolet or visible light absorbing groups in the reactive dye molecules [42], indicating that the paraffin@SiO_2_ colored microcapsules have excellent UV protection quality. Consequently, SiO_2_ shell and reactive dyes can protect ordinary building interior wall paints from UV, which is beneficial to improve its light aging resistance though absorbing UV.

Thermal stability is an important indicator for evaluating whether the microcapsule PCMs can withstand thermal cycles during operating, and it determines the practicability and durability of the microcapsule PCMs. It can be seen from Figure 11b that pristine paraffin has only experienced one weight loss within the heating temperature range, and the initial weight loss temperature is around 120 °C. When the temperature rises to 207 °C, paraffin will be completely thermally degraded. Simultaneously, the decomposition temperature of colored microcapsule PCMs increased to 148 °C, which indicates that due to the existence of stable inorganic rigid material SiO_2_, it can effectively prevent the thermal decomposition of core paraffin and increase the decomposition temperature of microcapsules. At the same time, the hydroxyl groups on the nano-SiO_2_ particles are thermally decomposed, causing a small amount of mass loss in the entire temperature range. As a result, it can be considered that the first mass loss in the colored microcapsules is mainly caused by the volatilization of the paraffin from the shell. Through calculation, it is concluded that the encapsulation rate of the colored microcapsules is 30%, which is consistent with the DSC results. More specifically, in the heating range from 230 °C to 600 °C, the colored microcapsules slowly lose their quality due to the decomposition of KH550 and reactive dyes and continue at 600 °C.

The infrared thermal images of the three samples from a to h in Figure 11c reveal a stable appearance characteristic of the microcapsules during the temperature rise. As the temperature rises, the paraffin sample gradually melts and eventually disappears entirely because of flowing away. The other two microcapsule samples still maintained a stable shape. This demonstrates that the microencapsulated core-shell structure can effectively prevent paraffin from leaking through melting so that it can be more effectively applied in thermal insulation systems.

### 3.8. Temperature Control Performance Testing of Internal Wall Insulation Thermal Coating

The thermal regulation capability was further studied by testing the two paints on a hot plate with infrared thermography; the sample is shown in Figure 12. The resulting infrared thermal image is shown in Figure 13. To ensure the same heat transfer rate, sample A is a normal colored latex paint coating prepared by adding colored SiO_2_ mixing with reactive dyes to normal latex paint coating, and sample B is a new type of colored latex paint coating containing paraffin@SiO_2_ colored microcapsules. An infrared thermal imager was employed to detect sample A and sample B simultaneously, and record once every 2 min. The resulting thermal images is displayed in Figure 13, in which can be observed the infrared thermal image of sample A over time, where the coating on the surface of the heat sink heats up at a constant temperature rise of a faster rate. When it is close to 12 min, the temperature of sample A has reached about 39.2 °C. On the contrary, the rising speed of the surface temperature of the glass substrate heat sink of sample B decreased significantly until the surface temperature of 18 min was 39.4 °C, which was about 2.5 °C lower than the surface temperature of the same period of 22 min. The experimental results indicated that the heat storage and release effect of PCMs can effectively reduce the heating rate and the final equilibrium temperature of the inner wall of the building.

It can be found that sample B containing PCM has obvious advantages in thermostat through comparing the infrared thermal analysis diagrams of the two sets of coatings, and it is feasible to further incorporate new types of colored microcapsules PCMs into building materials to improve energy issues.

## 4. Conclusions

In this paper, novel paraffin@SiO_2_ colored microcapsules that can be applied in interior wall coating are designed and manufactured through interfacial polymerization and chemical grafting methods. The observations of surface morphology demonstrated that the colored microcapsules had a regular spherical morphology and a well-defined core-shell structure. FT-IR and XRD spectrogram indicated that the resulting microcapsules contained SiO_2_ and reactive dyes. Thermal analysis showed that the obtained microcapsules reached a high phase-change enthalpy of more than 60 J/g and appropriate thermal energy storage and conversion efficiency of more than 30%, and the thermal conductivity was enhanced from 0.2649 to 0.7011 W/(m·K) after microencapsulation of paraffin with the colored SiO_2_ shell. TGA examination illustrated that the colored microcapsule PCMs possessed good thermal reliability and thermal stability. Consequently, the colored microcapsules exhibit good thermal energy storage capability and phase transition stability. Moreover, infrared thermal imaging analysis showed that the prepared colored latex paint coating containing paraffin@SiO_2_ colored microcapsules had an apparent ability to adjust temperature. Thus, it is feasible to apply paraffin@SiO_2_ colored microcapsules to building materials to alleviate energy problems and increase thermal comfort. However, the application of coloured phase-change microcapsules in architecture is still in the stage of experimental exploration. In order to better show its macroscopic temperature regulation effect, one can try to simulate the macroscopic device of the building to test and supplement the phase-change of colored microcapsules. In the future work, in-depth research should be conducted on the porosity, water absorption, compressive strength and other indicators of building wall materials.

## Figures and Tables

**Figure 1 materials-14-04012-f001:**
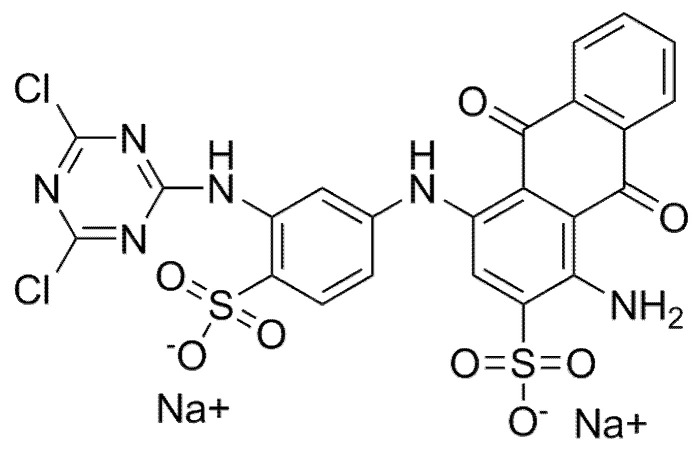
The structural formulas of a reactive dye (C.I. Reactive Blue 4 (X-BR).

**Figure 2 materials-14-04012-f002:**
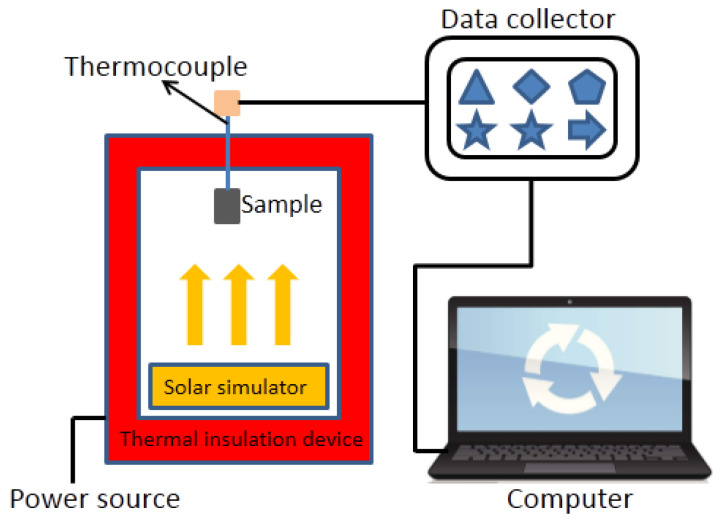
Time/temperature curve test device.

**Figure 3 materials-14-04012-f003:**
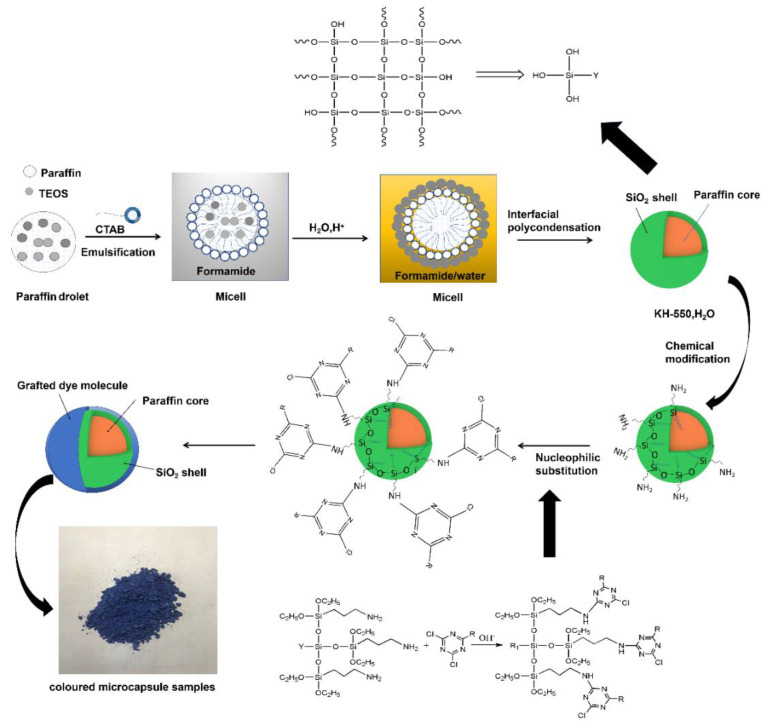
Synthetic strategy mechanism of colored microcapsules based on paraffin and colored SiO_2_ shell.

**Figure 4 materials-14-04012-f004:**
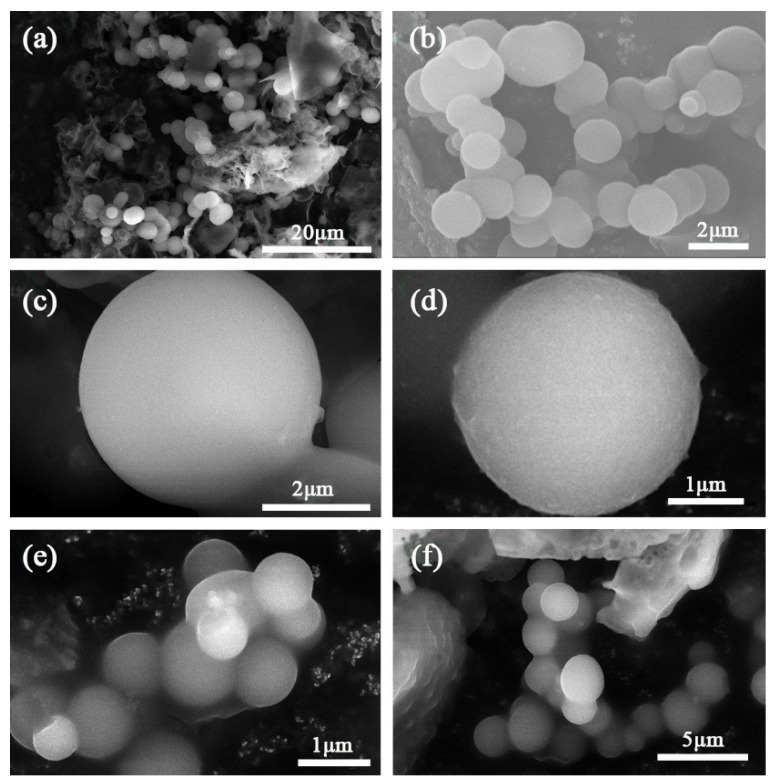
SEM micrographs of (**a**–**c**) paraffin@SiO_2_ microcapsules, (**d**–**f**) the paraffin@SiO_2_ colored microcapsules.

**Figure 5 materials-14-04012-f005:**
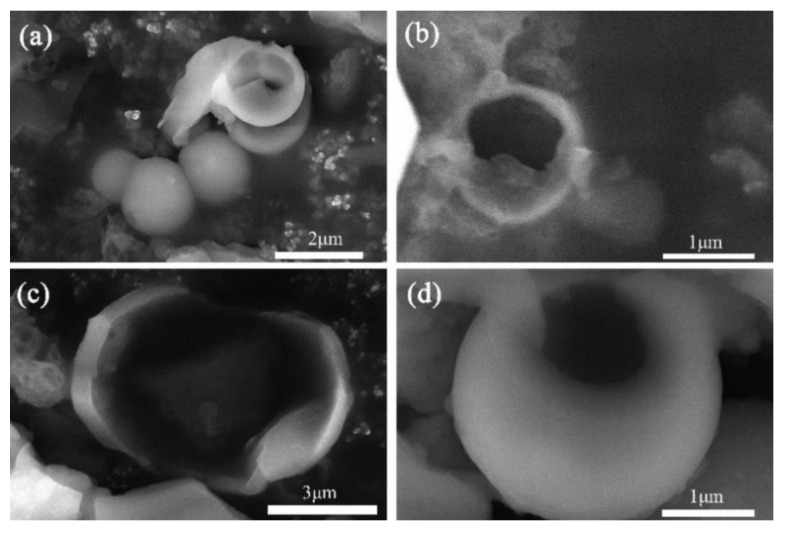
SEM micrographs of the deliberately damaged broken (**a**,**b**) paraffin@SiO_2_ microcapsules, (**c**,**d**) the paraffin@SiO_2_ colored microcapsules.

**Figure 6 materials-14-04012-f006:**
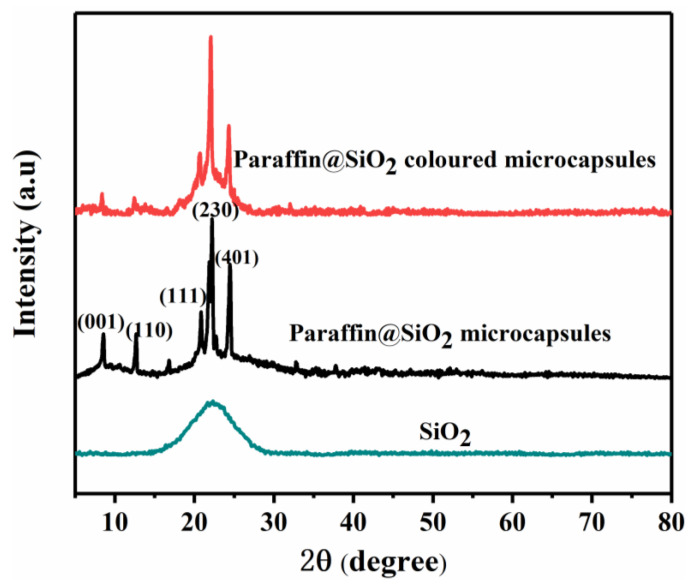
XRD patterns of the paraffin@SiO_2_ microcapsules, the paraffin@SiO_2_ colored microcapsules, and SiO_2_ nanoparticles.

**Figure 7 materials-14-04012-f007:**
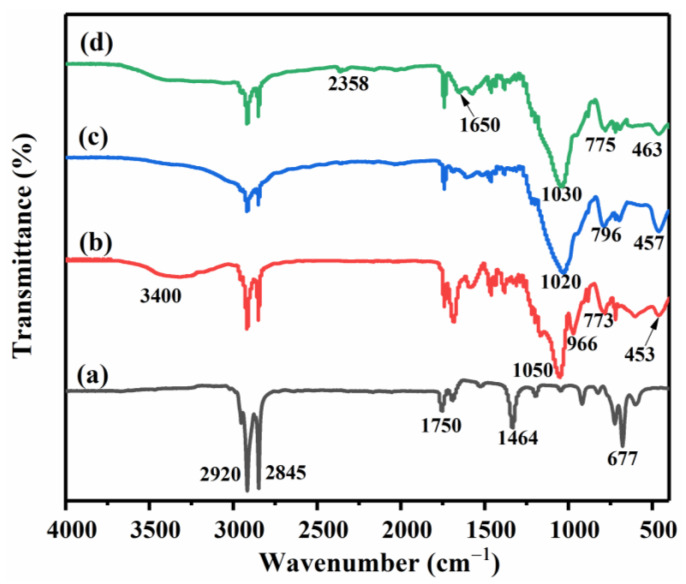
Infrared absorption spectrum of pristine paraffin (**a**), paraffin@SiO_2_ microcapsules (**b**), paraffin@SiO_2_ modified microcapsules (**c**), and paraffin@SiO_2_ colored microcapsules (**d**).

**Figure 8 materials-14-04012-f008:**
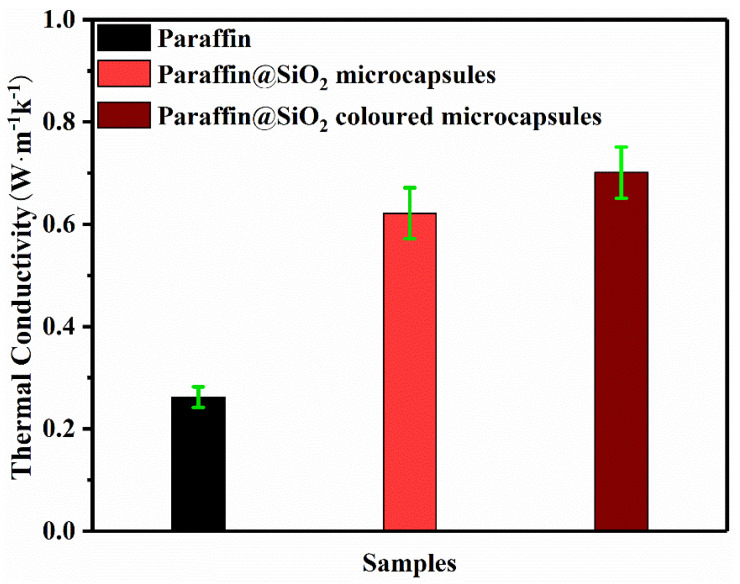
Thermal conductivity of pristine paraffin, paraffin@SiO_2_ microcapsules, the paraffin@SiO_2_ colored microcapsules, and SiO_2_ nanoparticles.

**Figure 9 materials-14-04012-f009:**
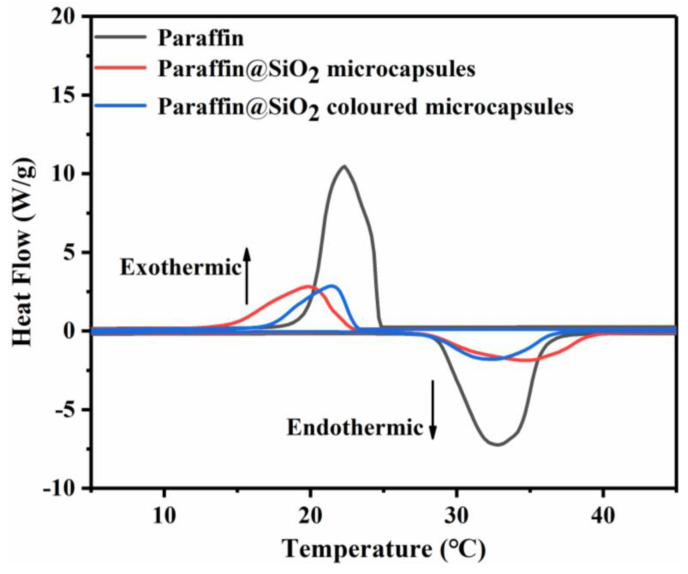
DSC of pristine paraffin, paraffin@SiO_2_ microcapsules, the paraffin@SiO_2_ colored microcapsules.

**Figure 10 materials-14-04012-f010:**
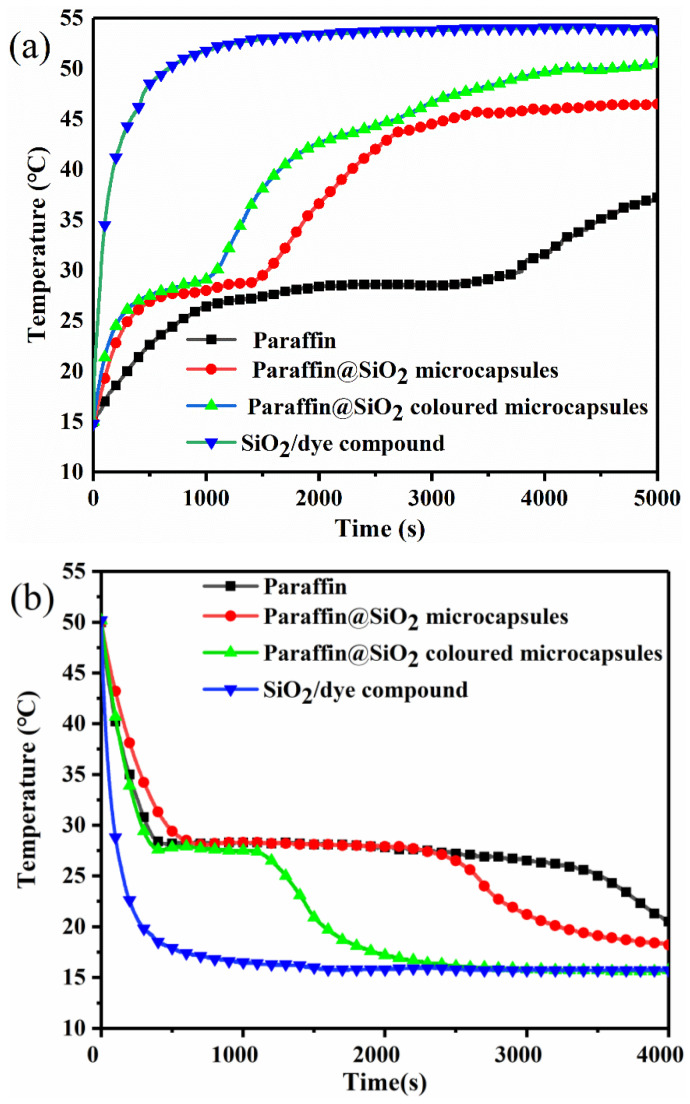
The temperature/time curves (**a**) under the simulated solar irradiance of 500 W·m^−2^ and (**b**) natural cooling at ambient temperature of 15 °C of pristine paraffin, paraffin@SiO_2_ microcapsules, the paraffin@SiO_2_ colored microcapsules, and SiO_2_ nanoparticles.

**Figure 11 materials-14-04012-f011:**
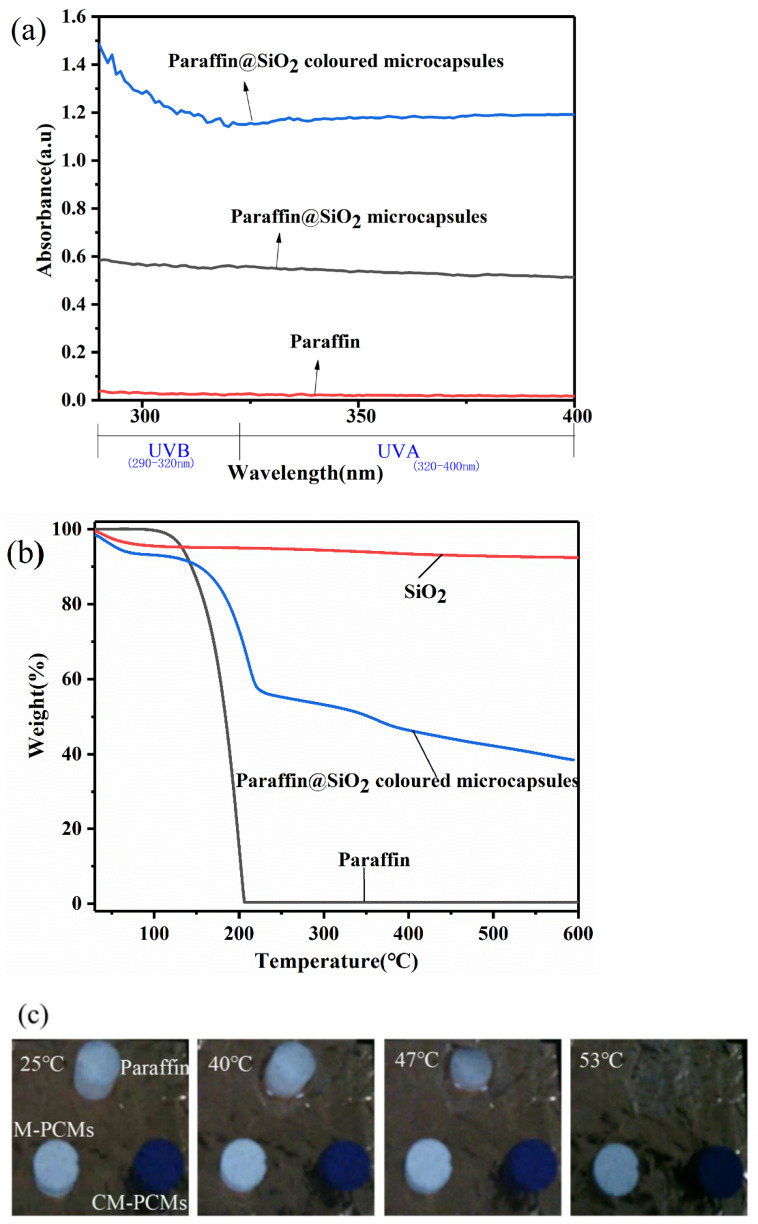
(**a**) UV–vis spectra of pristine paraffin, paraffin@SiO_2_ microcapsules, the paraffin@SiO_2_ colored microcapsules with same mass fraction; (**b**) TGA analysis of pristine paraffin, the paraffin@SiO_2_ colored microcapsules, and SiO_2_ nanoparticles; and (**c**) pictures before and after the leak test of the paraffin, paraffin@SiO_2_ microcapsules.

**Figure 12 materials-14-04012-f012:**
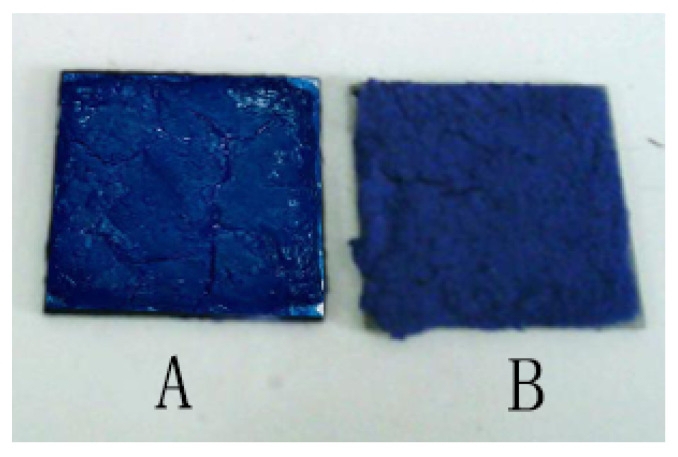
The digital photos of sample A (normal colored latex paint coating and sample B (novel colored latex paint coating containing paraffin@SiO_2_ colored microcapsules).

**Figure 13 materials-14-04012-f013:**
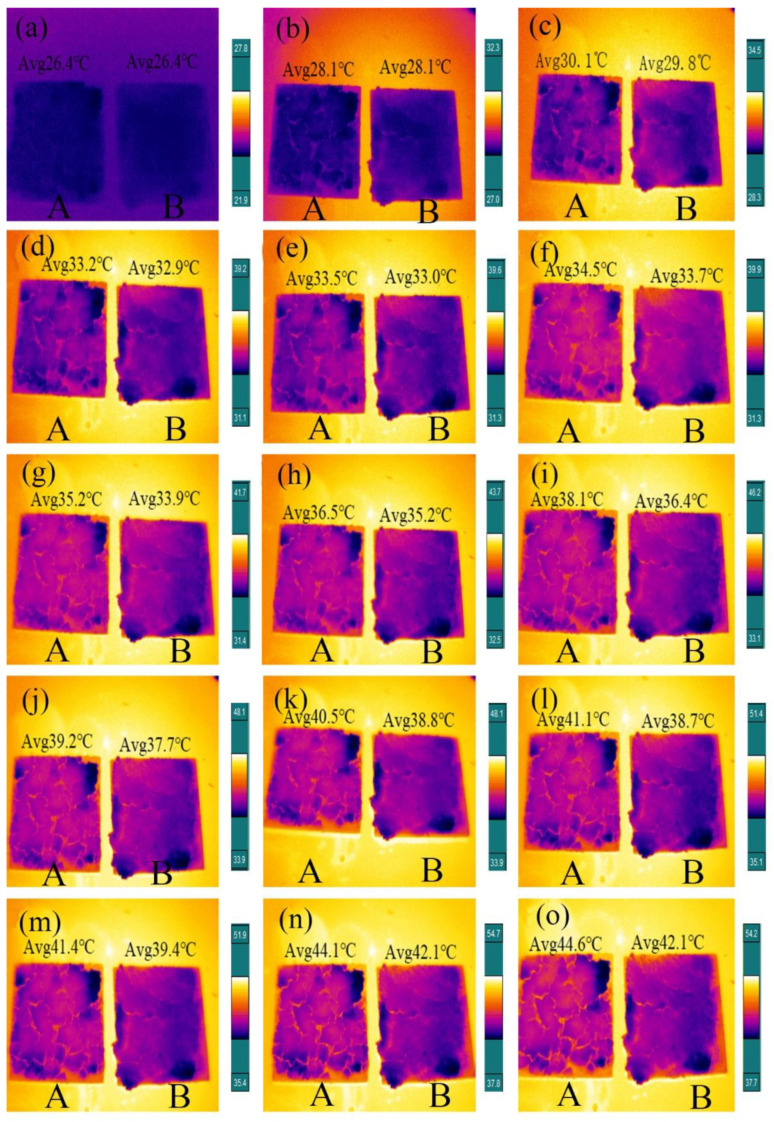
Thermal images (**a**–**o**) of sample A (normal colored latex paint coating) and sample B (novel colored latex paint coating containing paraffin@SiO_2_ colored microcapsules).

**Table 1 materials-14-04012-t001:** Properties of paraffin RT28HC used for research.

Parameter	Phase State	Result
Melting area (°C)	-	26–30
solidification area (°C)	-	27–23
Density (g/cm^3^)	Solid phase	0.862
	Liquid phase	0.850
Specific Heat capacity (kJ/Kg·K)	Solid phase	2.14
	Liquid phase	2.02
Thermal Conductivity (W/m·K)	Solid phase	0.26
	Liquid phase	0.25

**Table 2 materials-14-04012-t002:** Detailed preparation parameters for paraffin@SiO_2_ colored microcapsules.

Samples	Preparation Process	Temperature (°C)	Reaction Time (h)	Stirring Speed (r/min)
Paraffin	Paraffin/TEOS	45	0.5	1000
Microcapsules	Micelle	45	0.5	1000
	SiO_2_ shell	50	5	500
Modified microcapsule	KH550	40	2	500
Colored microcapsules	Colored SiO_2_ shell	40	2	300

**Table 3 materials-14-04012-t003:** Laboratory equipment used to perform the tests.

Type of Equipment	Parameter	Model and Company	Repetitions
Dryer	Temperature	Shanghai Experimental Instrument General Factory (ZK-82B)	-
Device for testing the thermal conductivity	Thermal conductivity	Xia tech hot plate thermal conductivity meter (TC-3000)	5
Time/temperature curve test device	Time/temperature curve	Self-assembled test device	3
differential scanning calorimeter	Thermal performance	American Waters Corporation(Q20)	-

**Table 4 materials-14-04012-t004:** Phase change properties of pristine paraffin and the obtained microcapsules.

Samples	Crystallization Process		Melting Process		Encapsulation Efficiency (%)	Thermal Conductivity (W·m^−1^·K^−1^)
	T_c_ (°C)	ΔH_c_ (J/g)	T_m_ (°C)	ΔH_m_ (J/g)		
Paraffin	28.56	203.49	24.98	206.15	-	0.2619
Microcapsules	27.93	77.31	22.48	78.23	38	0.6215
Colored microcapsules	27.98	60.49	23.16	59.03	30	0.7011

**Table 5 materials-14-04012-t005:** Color parameters of colored microcapsules.

Sample Code	Color Parameters				
	L*	a*	b*	c*	h*
Microcapsules	98.45	−0.08	−0.14	0.16	241.41
Colored microcapsules	54.71	−6.40	−25.30	26.09	255.81
X-Br	63.76	−3.23	−38.76	38.89	265.24

## Data Availability

Data is contained within the article.

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
