# Peer review of "Preparation of Colored Microcapsule Phase Change Materials with Colored SiO2 Shell for Thermal Energy Storage and Their Application in Latex Paint Coating"

_materials, 2021, doi:10.3390/ma14144012_

Round 1
Reviewer 1 Report
The paper presented is interesting and presents a relevant advancement in the fabrication of new materials with important thermal properties.
Overall, the paper has no particular issues, just minor remarks to be pointed out:
- Please, discuss about the safety and security for humans of the PCMs fabricated
- Please, add the main limitations of the products developed
Overall, several typos are present throughout the manuscript. Please, check and correct.
Author Response
Detailed Response to Reviewer
Dear Editor and Reviewer:
Thank you for your letter and for the reviewer’s comments concerning our manuscript entitled “Preparation of coloured microcapsule phase change materials with coloured SiO2 shell for thermal energy storage and their application in latex paint coating” (Manuscript No.: materials-1267391). Those comments are all valuable and very helpful for revising and improving our paper, as well as the important guiding significance to our researches. We have studied comments carefully and have made correction which we hope meet with approval. Revised portion are marked in yellow in the paper. The main corrections in the paper and the responds to the reviewer’s comments are as flowing:
1.Please, discuss about the safety and security for humans of the PCMs fabricated
Response: We appreciate the careful reading of our manuscript and valuable question of the reviewer very much. Your opinion is very constructive, due to time reasons, this will be the research direction of our next topic.
2.Please, add the main limitations of the products developed
Response: We appreciate the careful reading of our manuscript and valuable question of the reviewer very much. After testing and analysis, it was found that the coloured microcapsules not only showed good thermal physical properties, thermal conductivity and thermal stability, but also showed bright colours. Therefore, the application of paraffin@SiO2 coloured microcapsules to building materials can not only alleviate energy problems and improve thermal comfort, but also greatly increase the colour diversity of interior wall coatings. However, the application of coloured phase-change microcapsules in architecture is still in the stage of experimental exploration. In order to better show its macroscopic temperature regulation effect, you can try to simulate the macroscopic device of the building to test and supplement the phase-change of coloured microcapsules. Material research on the porosity, water absorption, compressive strength and other indicators of building wall materials.
3.Overall, several typos are present throughout the manuscript. Please, check and correct.
Response: We appreciate the careful reading of our manuscript and valuable question of the reviewer very much. We are sorry for some formation/grammatical errors in the paper. For your suggestion, we have made corresponding changes to the article. For example:
- In addition, it can be acknowledged from public research that there are many organic or inorganic materials that can be used as solid-liquid PCM, including organic fatty acids and paraffins, inorganic salt hydrates and esters[4-6].
- Add containing 15% coloured SiO2 mixing with reactive dyes as a comparison and 15% paraffin@SiO2 coloured microcapsules to a latex paint coating formula (65% silicone acrylic emulsion, 1.5% ethylene glycol, Alcohol ester 2.0%, bactericide 0.2%, thickener 0.6%, water) are mixed evenly by magnetic stirring at 1000 rpm for half an hour to obtain normal coloured coating.
- At a heating rate of 10°C/min from room temperature to 700°C, and a flow rate of 50mL/min, thermal stability of the coloured microcapsules and paraffin samples was studied under nitro conditions applying a thermogravimetric analyzer (TGA, Q5000).
4.In addition, it is generally accepted that the latent heat of PCMs is one of the most significant factors affecting working performance evaluating the phase change behavior of the prepared microcapsule samples. The phase transition behavior of pristine paraffin and microcapsule samples can be investigated by DSC, and their DSC curve and related phase transition parameters are as follows.
- Figure 12. The digital photos of sample A (normal coloured latex paint coating and sample B (novel coloured latex paint coating containing paraffin@SiO2 coloured microcapsules).
- To ensure the same heat transfer rate, sample A is a normal coloured latex paint coating prepared by adding coloured SiO2 mixing with reactive dyes to normal latex paint coating, and sample B is a new type of coloured latex paint coating containing paraffin@SiO2 coloured microcapsules.
7.It can be found that sample B containing PCM has obvious advantages in thermostat through comparing the infrared thermal analysis diagrams of the two sets of coatings, and it is feasible to further incorporate new types of coloured microcapsules PCMs into building materials to improve energy issues.
For your suggestions, we have made corresponding changes to this article including tense, grammar, sentence structure, and the wrong places also made corrections. These changes, corrections and supplements are marked accordingly in the paper. We hope that the revised article will be able to meet the journal and your publishing requirements, and we thank the reviewers once again for both their positive reviews and the helpful comments and suggestions.

Reviewer 2 Report
The paper is aimed at experimental investigating the thermal energy storage properties of colored MPCMs, with SiO2 shell pigments and their application in latex paint coating. The work tackles an interesting topic and it is suitable for Materials J.
The work however needs the following improvements and clarifications before its approval. In the Reviewer's opinion the following recommendations/clarifications should be considered:
- In section 1, after the State of the Art (SoA), the Authors should clarify what are the key novelties of this paper and the main contributions of this work beyond the current SoA. They are not clearly stated.
- An overview of the Rubitherm 28 (probably the RT28HC®) should be mentioned through the text: i.e., melting/solidification area ranges, Heat storage capacity (latent and sensible), heat conductivities (in both phases) as declared in the datasheet can be reported.
- DSC tests were performed at almost high heating rate of 10°C/min and by considering a small mass amount: i.e. 7±1 mg of each sample. In my opinion, the heating rate was too high, which does not permit to representatively capture the melting and the solidification points correctly. They are actually affected by the non-reached thermal equilibrium.
- Heating rate issues, which can affect the DSC results in PCM in general, were not properly taking into account through the manuscript (or barely addressed). Please check the DSC procedure and measurements for PCM as developed in the IEA SHC TASK 42/ECES Task (or see also https://www.mdpi.com/1996-1944/13/7/1705).
- Between section 2 and 3, after presenting the materials (components and reagents), please summarize in a unique table the experimental program as whole for having a general overview of it. Thus, showing the number of specimens/repetitions per components (PCM, MPCM, colored MPMC), and the achievable/measurable parameters for each (melting/solidification temperatures, conductivities, etc.). This will help the Reader to quickly understand the experimental campaign as a whole.
- Please declare in which state (liquid, solid, both) were the conductivity measurements performed?
- Please show the scatters of the experimental results (I suppose that more than one specimen per test type/mixture were considered). This applies for all Figs. with histograms by adding +/- variabilities: error bars, or by showing the grey area of variability in common X-Y graphs.
- Quality of tables and figures should be improved.
Author Response
Detailed Response to Reviewer
Dear Editor and Reviewer:
Thank you for your letter and for the reviewer’s comments concerning our manuscript entitled “Preparation of coloured microcapsule phase change materials with coloured SiO2 shell for thermal energy storage and their application in latex paint coating” (Manuscript No.: materials-1267391). Those comments are all valuable and very helpful for revising and improving our paper, as well as the important guiding significance to our researches. We have studied comments carefully and have made correction which we hope meet with approval. Revised portion are marked in yellow in the paper. The main corrections in the paper and the responds to the reviewer’s comments are as flowing:
- In section 1, after the State of the Art (SoA), the Authors should clarify what are the key novelties of this paper and the main contributions of this work beyond the current SoA. They are not clearly stated.
Response: Thank you for the careful reading of our manuscript and the valuable suggestions of the reviewer very much. The innovation of this article is on the basis of the best morphology silica inorganic wall material microcapsules, the active hydroxyl groups on the surface of the paraffin@SiO2 phase change microcapsules are modified with a silane coupling agent through triaminopropyltriethylsilane (KH550). Thus, synthesize organic modified paraffin @SiO2 phase change microcapsules containing amino groups on the surface. Then carry out a nucleophilic substitution reaction between the introduced amino group and the active Cl atom of the organic reactive dye. After testing and analysis, it was found that the coloured microcapsules not only showed good thermal physical properties, thermal conductivity and thermal stability, but also showed bright colors. Therefore, the application of paraffin@SiO2 coloured microcapsules to building materials can not only alleviate energy problems and improve thermal comfort, but also greatly increase the colour diversity of interior wall coatings.
- An overview of the Rubitherm 28 (probably the RT28HC®) should be mentioned through the text: i.e., melting/solidification area ranges, Heat storage capacity (latent and sensible), heat conductivities (in both phases) as declared in the datasheet can be reported.
Response: We appreciate the careful reading of our manuscript and valuable question of the reviewer very much. According to your suggestions, we have added the relevant parameters of RT28HC®, as shown in the following table
Table 1. Properties of paraffin RT28HC used for research
Parameter |
Phase state |
Result |
Melting area (℃) |
- |
26-30 |
solidification area (℃) |
- |
27-23 |
Density (g/cm3) |
Solid phase |
0.862 |
Liquid phase |
0.850 |
|
Specific Heat capacity (kJ/Kg·K) |
Solid phase |
2.140 |
Liquid phase |
2.020 |
|
Thermal Conductivity (W/m·K) |
Solid phase |
0.260 |
Liquid phase |
0.250 |
- DSC tests were performed at almost high heating rate of 10°C/min and by considering a small mass amount: i.e. 7±1 mg of each sample. In my opinion, the heating rate was too high, which does not permit to representatively capture the melting and the solidification points correctly. They are actually affected by the non-reached thermal equilibrium.
Response: We appreciate the careful reading of our manuscript and valuable question of the reviewer very much. Your suggestion is very useful. Unfortunately, there is a problem with our testing equipment and it is under repair. In addition, we selected the heating rate according to the literature on phase change microcapsules. For example:
- Zhang et al [1] recorded by DSC scans on a TA Instruments Q100 differential scanning calorimeter under a nitrogen atmosphere at a heating or cooling rate of 10 ℃/min.
- Liu et al [2]measured the phase-change performance of microcapsule specimens by differential scanning calorimetry (DSC) using a differential scanning calorimeter (TA Instruments Q20) at a scanning rate of 10 °C/min.
- Chai et al [3] analyze the phase change characteristics of the microcapsule samples on a TA Instruments Q20 differential scanning calorimeter equipped with a thermal analysis data station. All of the measurements were carried out under a nitrogen atmosphere at a heating or cooling rate of 10 ℃/min, and the mass for each specimen is about 5–6 mg. The first heating scan was run from 20 to 60 C, and the specimens were held at this temperature for 3 min to diminish the thermal history before the formal measurement.
- Nihal Sarier et al [4] conducted DSC analyses on a Perkin-Elmer/Pyris 1 type DSC under nitrogen atmosphere to characterize the thermal behavior of PCMs and their microcapsules, which consist of polymer shells. During DSC analyses, test specimens were heated and cooled within a certain temperature interval ranging from 0 ◦C to 55 ℃ at 10 ℃·min−1, which is commonly used in the experiments of polymer microcapsules
- Xu et al [5] studied the thermal properties of paraffin@Cu-Cu2O/CNTs microcapsules by differential scanning calorimetry under nitrogen condition using DSC (STARe System, Germany) with a 10 °C/min ramp between 10 and 100 °C to obtain thermal parameters such as phase change temperature and latent heat of paraffin@Cu-Cu2O/CNTs microcapsules and paraffin.
[1] Li B, Liu T, Hu L, et al. Fabrication and Properties of Microencapsulated Paraffin@SiO2 Phase Change Composite for Thermal Energy Storage[J]. Acs Sustainable Chemistry & Engineering, 2013, 1(3):374-380.
[2] H. Liu, X. Wang, D. Wu, Tailoring of bifunctional microencapsulated phase change materials with CdS/SiO2 double-layered shell for solar photocatalysis and solar thermal energy storage, Applied Thermal Engineering, 134 (2018) 603-614.
[3] Chai L, Wang X , D Wu. Development of bifunctional microencapsulated phase change materials with crystalline titanium dioxide shell for latent-heat storage and photocatalytic effectiveness[J]. Applied Energy, 2015, 138(jan.15):661–674.
[4] Sarier N, Onder E . The manufacture of microencapsulated phase change materials suitable for the design of thermally enhanced fabrics[J]. Thermochimica Acta, 2007, 452(2):149-160.
[5] Xu B, Chen C, J Zhou, et al. Preparation of novel microencapsulated phase change material with Cu-Cu2O/CNTs as the shell and their dispersed slurry for direct absorption solar collectors[J]. Solar Energy Materials and Solar Cells, 2019, 200:109980-.
- Heating rate issues, which can affect the DSC results in PCM in general, were not properly taking into account through the manuscript (or barely addressed). Please check the DSC procedure and measurements for PCM as developed in the IEA SHC TASK 42/ECES Task (or see also https://www.mdpi.com/1996-1944/13/7/1705).
Response: We appreciate the careful reading of our manuscript and valuable question of the reviewer very much. For your question, we make the following explanation. Reply the same as 3
- Between section 2 and 3, after presenting the materials (components and reagents), please summarize in a unique table the experimental program as whole for having a general overview of it. Thus, showing the number of specimens/repetitions per components (PCM, MPCM, coloured MPMC), and the achievable/measurable parameters for each (melting/solidification temperatures, conductivities, etc.). This will help the Reader to quickly understand the experimental campaign as a whole.
Response: We appreciate the careful reading of our manuscript and valuable question of the reviewer very much. According to your suggestions, we have added the relevant detailed preparation parameters for paraffin@SiO2 coloured microcapsules, as shown in the following table:
Table 2 Detailed preparation parameters for paraffin@SiO2 coloured microcapsules
Samples |
Preparation process |
Temperature (℃) |
Reaction time (h) |
Stirring speed (r/min) |
|
Paraffin |
Paraffin/TEOS |
45 |
0.5 |
1000 |
|
Microcapsules |
Micelle |
45 |
0.5 |
1000 |
|
SiO2 shell |
50 |
5 |
500 |
||
Modified microcapsule |
KH550 |
40 |
2 |
500 |
|
Coloued microcapsules |
Coloured SiO2 shell |
40 |
2 |
300 |
|
Table 3. Laboratory equipment used to perform the tests.
Type of Equipment |
Parameter |
Model and Company |
Repetitions |
Dryer |
Temperature |
Shanghai Experimental Instrument General Factory (ZK-82B) |
- |
Device for testing the thermal conductivity |
Thermal conductivity |
Xia tech hot plate thermal conductivity meter (TC-3000) |
5 |
Time/temperature curve test device |
Time/temperature curve |
self-assembled test device |
3 |
differential scanning calorimeter |
Thermal performance |
American Waters Corporation(Q20) |
- |
- Please declare in which state (liquid, solid, both) were the conductivity measurements performed?
Response: We appreciate the careful reading of our manuscript and valuable question of the reviewer very much. In this article, the thermal conductivity is measured at the current room temperature (26°C)
- Please show the scatters of the experimental results (I suppose that more than one specimen per test type/mixture were considered). This applies for all Figs. with histograms by adding +/- variabilities: error bars, or by showing the grey area of variability in common X-Y graphs.
Response: We appreciate the careful reading of our manuscript and valuable question of the reviewer very much. According to your suggestion, we have added a variable area (error bar) to the relevant image, as shown in the figure below:
Figure 8. Thermal conductivity of pristine paraffin, paraffin@SiO2 microcapsules, the paraffin@SiO2 coloured microcapsules and SiO2 nanopaticles.
- Quality of tables and figures should be improved.
Response: We appreciate the careful reading of our manuscript and valuable question of the reviewer very much. We carefully checked the figures and tables in the article according to your opinions and made corresponding changes in the article. For example:
- As shown in Figure 4, Figure5 and Figure 11(c), we changed the colour of the SEM image annotation font to white and unified it into English half-width, and then adjusted the image size.
Figure 4. SEM micrographs of (a–c) paraffin@SiO2 microcapsules, (d–f) the paraffin@SiO2 coloured microcapsules.
Figure 5. SEM micrographs of the deliberately damaged broken (a,b) paraffin@SiO2 microcapsules, (c,d) the paraffin@SiO2 colored microcapsules.
Figure 11. (a) UV–vis spectra of pristine paraffin, paraffin@SiO2 microcapsules, the paraffin@SiO2 coloured microcapsules with same mass fraction, (b) TGA analysis of pristine paraffin, the paraffin@SiO2 coloured microcapsules and SiO2 nanopaticles and (c) pictures before and after the leak test of the paraffin, paraffin@SiO2 microcapsules
- As shown in Table 4, we adjusted the overall format of the three-line table and added a column for thermal conductivity.
Table 4. Phase change properties of pristine paraffin and the obtained microcapsules.
Samples |
Crystallization process |
Melting process |
Encapsulation efficiency (%) |
Thermal conductivity (W·m−1·K−1) |
|||
Tc (℃) |
ΔHc (J/g) |
Tm (℃) |
ΔHm (J/g) |
||||
Paraffin |
28.56 |
203.49 |
24.98 |
206.15 |
- |
0.2619 |
|
Microcapsules |
27.93 |
77.31 |
22.48 |
78.23 |
38 |
0.6215 |
|
Coloured microcapsules |
27.98 |
60.49 |
23.16 |
59.03 |
30 |
0.7011 |
|
- As shown in Table 12 and Table 13, we have unified the English half-width and brackets in the figure, and made corresponding adjustments.
Figure 12. The digital photos of sample A (normal coloured latex paint coating and sample B (novel coloured latex paint coating containing paraffin@SiO2 coloured microcapsules).
Figure 13. Thermal images(a-o)of sample A (normal coloured latex paint coating) and sample B (novel coloured latex paint coating containing paraffin@SiO2 coloured microcapsules).
In addition, for all graphs, we modified the size of the annotations and coordinate annotations in the graphs, and adjusted the width of the curves to make the images easier to identify and better show to readers. We hope that the revised article will be able to meet the journal and your publishing requirements, and we thank the reviewers once again for both their positive reviews and the helpful comments and suggestions.

Round 2
Reviewer 2 Report
The paper can be now accepted